# Oxygen and Pt(II) self-generating conjugate for synergistic photo-chemo therapy of hypoxic tumor

Shuting Xu[1], Xinyuan Zhu[1], Chuan Zhang [1], Wei Huang[1], Yongfeng Zhou [1] & Deyue Yan[1]

Cancer cells in hypoxic tumors are remarkably resistant to photodynamic therapy. Here, we hypothesize that an oxygen and Pt(II) self-generating multifunctional nanocomposite could reverse the hypoxia-triggered PDT resistance. The nanocomposite contains Pt(IV) and chlorin e6, in which upconversion nanoparticles are loaded to convert 980 nm near-infrared light into 365 nm and 660 nm emissions. Upon accumulation at the tumor site, a 980 nm laser is used to trigger the nanocomposite to generate $O_2$ for consumption in the PDT process and to produce cytotoxic reactive oxygen species. The composite also releases active Pt(II) for synergistic photo-chemo therapy to enhance antitumor efficiency. The oxygen and Pt(II) self-generating prodrug is shown to have high potential to inhibit tumors out of the range of UV light, to overcome the hypoxia-triggered PDT resistance and significantly improve anticancer efficacy by the synergistic PDT-chemotherapy.

[1] School of Chemistry and Chemical Engineering, State Key Laboratory of Metal Matrix Composites, Shanghai Jiao Tong University, 800 Dongchuan Road, Shanghai 200240, P. R. China. Correspondence and requests for materials should be addressed to D.Y. (email: dyyan@sjtu.edu.cn)

Hypoxia, a key feature of solid tumor microenvironment due to malignant cancer cell proliferation and blood vessel deformation during tumor angiogenesis[1,2], is found to be responsible for the resistance of many therapeutic approaches, such as chemotherapy, radiotherapy, and photodynamic therapy (PDT)[1,3–7]. Particularly, oxygen would be significantly consumed during the PDT process to generate reactive oxygen species (ROS), thus the existence of hypoxia usually leads to low efficacy of PDT treatment[8].

To overcome this problem, different strategies have been developed to alleviate tumor hypoxia during the PDT treatment. For instance, in a hyperbaric oxygen (HBO) therapy, pure oxygen is offered to patient in a pressurized sealed chamber to promote oxygen transport to the hypoxic regions of tumor[9]. Unfortunately, side effects, including barotrauma and hyperoxic seizures, greatly limit its application in clinic[10–13]. Alternatively, various $O_2$-generating materials including $MnO_2$[14,15], catalase[16,17], and perfluorohexane[18,19] have been used to overcome hypoxia in recent reports. However, $MnO_2$ and catalase highly rely on catalyzing the endogenous $H_2O_2$ to evolve $O_2$ and perfluorocarbon-based oxygen carriers normally transport limited oxygen to the tumor tissue, all of which exhibit moderate capabilities on reducing intratumoral hypoxia[20]. This work manages to design a drug complex that can generate oxygen to meet the demands of PDT process and alleviate the hypoxia simultaneously, while producing cytotoxic chemotherapeutics for a synergistic cancer therapy.

To achieve this aim, a series of platinum(IV)-azide complexes were found to be ideal candidates, which are inert in dark and even in the existence of millimolar glutathione[21], but sensitive to light[22–25]. Importantly, platinum(IV) diazido complexes containing cis-diammine ligands can be triggered to evolve gaseous dioxygen and generate active cytotoxic Pt(II) simultaneously under light irradiation[22,26]. Therefore, we speculate that upon incorporating platinum(IV) diazido complexes in a conventional PDT treatment, the photosensitizers can utilize the locally-generated oxygen to produce ROS. At the meantime, the photo-generated Pt(II) complex could serve as cytotoxic chemotherapeutics to kill the tumor cells and enhance the overall antitumor effect in a synergistic manner. Unfortunately, most of the photoactive platinum(IV) diazido complexes only respond to considerably short wavelength light (e.g. UV or blue light), which is not favorable in the PDT treatment owing to the unexpected photodamage to normal tissues and short light/tissue penetration. It is expected that the drug complex would be activated under only one long wavelength light for the combined PDT-chemotherapy to match the conventional PDT module and optimize its potential for real clinical applications.

Herein, we demonstrate a near infrared (NIR) light-controlled and $O_2$/Pt(II) self-generating prodrug that can enhance PDT efficacy for a synergistic photo-chemo antitumor therapy. Briefly, the anticancer complex cis,trans,cis-[Pt(N₃)₂(OH)₂(NH₃)₂] (denoted as Pt(IV)) is selected as a promising $O_2$-self generating material in the PDT therapy actuated by photosensitizer chlorin e6 (Ce6). The amphiphilic oligomer Ce6-PEG-Pt(IV) (CPP) can self-assemble into micelles (Fig. 1b). Because of the limited decomposition of platinum(IV) diazido complexes under long wavelength light, upconversion nanoparticles (UCNPs) NaYbF₄:Tm@CaF₂ are co-assembled with CPP to obtain UCNPs-embedded nanoparticles (denoted as UCPP). At the meantime, UCNPs-embedded PEG-Ce6 nanoparticles (denoted as UPC) and UCNPs-loaded PEG-Pt(IV) nanoparticles (denoted as UPP) are used as control. Notably, the UCNPs can absorb 980 nm NIR light and convert it into 365 nm and 660 nm emissions, which will induce the decomposition of Pt(IV) and drive PDT in UCPP opportunely (Fig. 1b, c). When a 980 nm laser irradiation is applied to trigger UCPP decomposition, $O_2$ can be generated to compensate the consuming of oxygen during the PDT process and active Pt(II) can be released for a synergistic photo-chemo therapy (Fig. 1a). In this approach, co-delivery of $O_2$-evolving materials, such as $MnO_2$ or catalase etc., is not necessary and the restriction of hypoxia during the PDT treatment can be addressed. Associated with photo-generated Pt(II) species, anticancer efficacy can be dramatically enhanced by the synergistic PDT-chemotherapy.

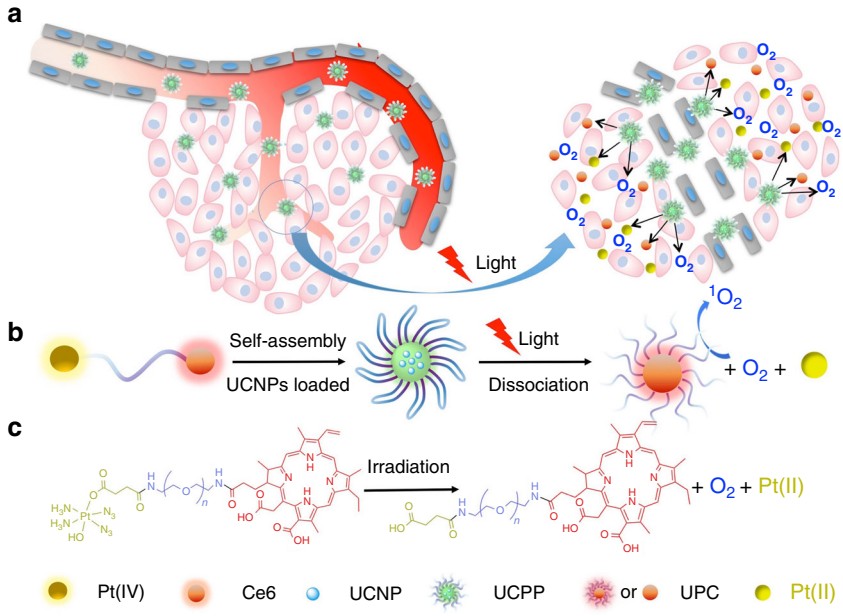

**Fig. 1** Schematic illustration of UCPP nanoparticle. **a** A scheme illustrating the light-driven dissociation of UCPP in tumor microenvironment. **b** Formation of UCPP and its light-driven dissociation including $O_2$ self-generation, simultaneous activation of PDT, and active Pt(II) releasing for the synergistic photo-chemotherapy. **c** The chemical dissociation mechanism of UCPP (for details refer to ref. [27])

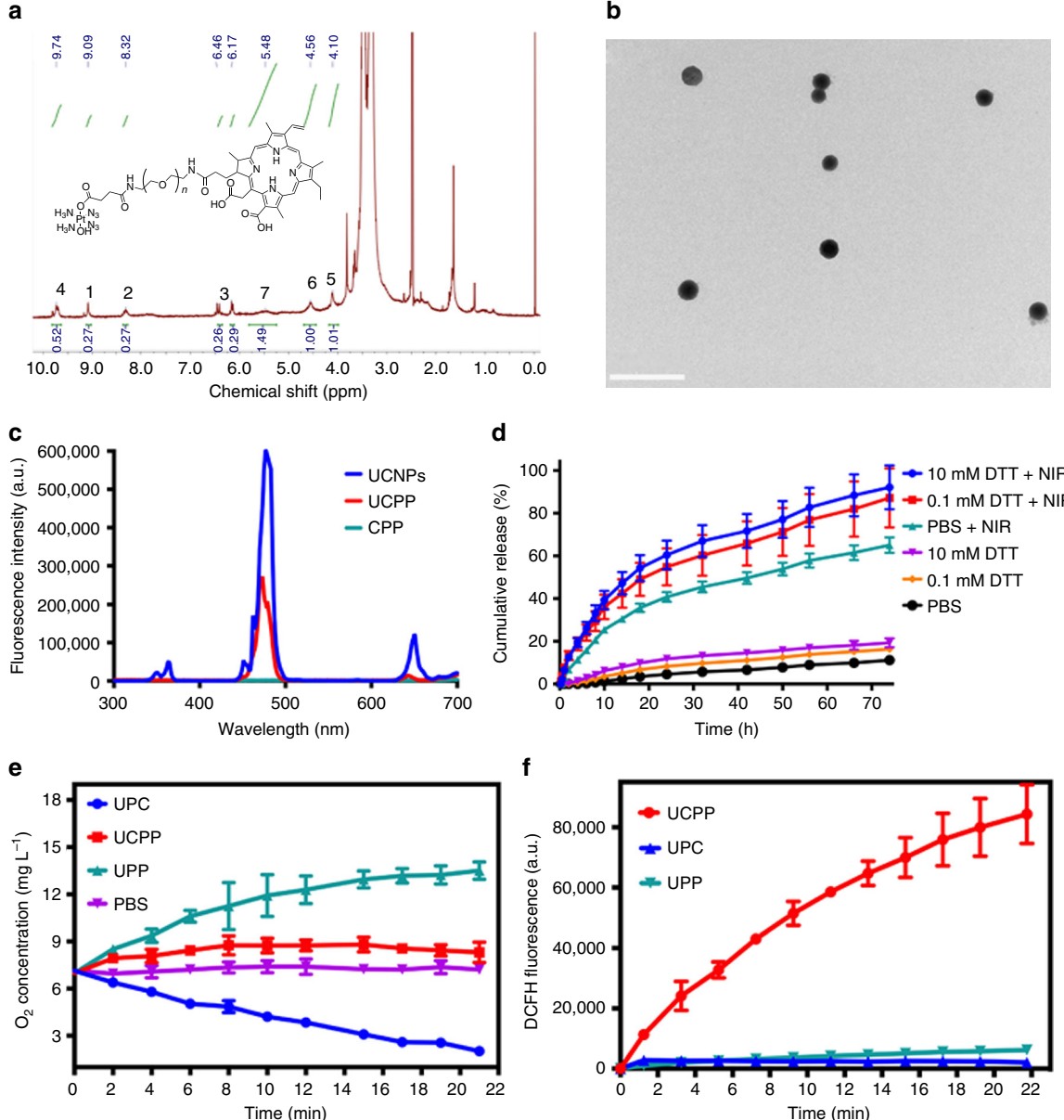

**Fig. 2** Characterization of UCPP. **a** $^1$H NMR spectrum of prodrug CPP; **b** TEM image of UCPP. Scale bar = 0.5 μm; **c** Emission spectra of pure UCNPs (blue line), UCPP (red line) and pure CPP (green line) under 980 nm laser; **d** Pt release from UCPP in different concentrations of DTT buffers under dark or NIR irradiation.; **e** Oxygen generation or consumption of UPP, UCPP and UPC under 980 nm irradiation (0.85 W cm$^{-2}$); **f** The generation of ROS of UCPP, UPP or UPC under 980 nm irradiation (0.85 W cm$^{-2}$) in hypoxia environment, determined by the fluorescence intensity of DCFH. Data were shown as mean ± S.D. ($n = 3$)

## Results

**Synthesis and characterization of UCPP.** UCPP nanoparticles were prepared through supramolecular self-assembly of the amphiphilic prodrug CPP together with UCNPs. The details of synthesis are presented in Supplementary Methods. The $^1$H nuclear magnetic resonance ($^1$H NMR) spectrum of CPP is shown in Fig. 2a. The area integral of the peak at 9.74 ppm (4) for Ce6 protons and that at 5.48 ppm (7) relating to ammine protons of Pt(IV) are nearly 0.5 and 1.5 respectively when the area integral of PEG-α-methylene protons (6) at 4.56 ppm is defined as 1. This result indicated that Pt(IV) and Ce6 were conjugated to PEG with the molar ratio of 1:1. Furthermore, the content of Pt measured by Elan DRC II inductively coupled plasma mass (ICP-MS) and that of Ce6 measured by UV-vis absorption spectroscopy are 6.5 wt% and 20.1 wt% respectively, which also confirms

the molar ratio of Pt(IV) and Ce6 is 1:1 (Supplementary Fig. 9 and Supplementary Discussion 2). $^1$H NMR, Fourier transform infrared (FTIR), UV-vis, and X-ray photoelectron spectroscopy (XPS) spectra further certified the successful synthesis of CPP prodrug and its control samples (Supplementary Figs 1–8 and Supplementary Discussion 1). On the other hand, UCNPs based on 99.5% Yb and 0.5% Tm-doped NaYbF$_4$:Tm@CaF$_2$ nanoparticles were synthesized according to the previously established method (Supplementary Fig. 10 and Supplementary Discussion 3)[27]. The amphiphilic CPP can co-self-assemble with UCNPs into UCPP nanoparticles in water. Transmission electron microscopy (TEM) and dynamic light scattering (DLS) are used to determine the morphology and size of UCPP nanoparticles. The TEM image in Fig. 2b displays the average size of UCPP nanoparticles is approximately 45 nm, which is smaller than that

measured by DLS (Dh = 60.7 nm) due to the shrinkage under dry state (Supplementary Fig. 11).

UCPP is a kind of complex nanoparticles consisting of UCNPs and CPP, in which the upconverted emissions from UCNPs can activate Pt(IV) and Ce6 of the CPP prodrug. The upconversion emission spectra of UCNPs, UCPP, and CPP are shown in Fig. 2c. Undergoing NIR irradiation, the CPP prodrug had no fluorescence emission, while UCNPs had upconversion emission peaks at 365 nm, 470 nm and 660 nm. But for UCPP, the upconversion emission intensities at 365 nm and 660 nm decreased sharply, which indicated they were absorbed by CPP. This result verifies that the upconverted emissions from UCNPs can trigger the decomposition of Pt(IV) and motivate the PDT process. Moreover, the blue emission of the UCNPs may motivate Pt(IV) as well[21]. Hence, the emission intensity of UCPP at 470 nm was much weaker than that of UCNPs. All above data confirmed the energy transfer process occurred between CPP and UCNPs.

Based on the stimulus-response behavior of UCPP to NIR irradiation, the drug release of UCPP were conducted in different buffer solutions in dark or upon NIR irradiation. As shown in Fig. 2d, UCPP displayed limited Pt release rates in phosphate-buffered saline (PBS), 0.1 mM or 10 mM DL-Dithiothreitol (DTT) buffer solutions in dark. Compared to conventional platinum(IV) complexes, which is easily reduced to platinum(II) by DTT, UCPP showed significantly stable before reaching the tumor site. Thus, undesirable side reactions with intracellular reducing agents or proteins can be avoided in principle by using UCPP. Interestingly, when UCPP was exposed to NIR light, the release of Pt were immediately boosted either in PBS or DTT buffer solutions. Such a site-selective photo-activation would lead to the decreased side effects and the increased therapeutic efficacy as a result.

**In vitro study of oxygen generating and PDT enhancement.** One of the fundamental arguments in this study is the oxygen producing capability of UCPP under irradiation. The generated oxygen of UCPP can be used to compensate the $O_2$ consuming during the PDT process and alleviate the hypoxia. Hence, $O_2$ and ROS generating abilities were evaluated by exposing UCPP to the NIR light in PBS. A dissolved oxygen meter was used to measure the $O_2$ concentration in the PBS buffers when the samples were exposed to 980 nm laser. As shown in Fig. 2e, upon exposing to the 980 nm laser, the $O_2$ concentration in UPP-dispersed solution was gradually increasing with the time and then reached a plateau as the decomposition of platinum(IV) approached completion. In contrast, the $O_2$ concentration in UPC-dispersed solution decreased because Ce6 consumed $O_2$ to generate singlet oxygen ($^1O_2$) under the upconverted emission at 660 nm from UCNPs. Since Ce6 in UCPP consumed most of $O_2$ produced by Pt(IV) in UCPP, the UCPP-dispersed solution showed a slower increase of oxygen concentration compared with that of the UPP-dispersed solution. In other words, the $O_2$ generated from UCPP is more than that consumed in motivating PDT. Furthermore, ROS generation was measured by the ROS sensor 2′,7′-

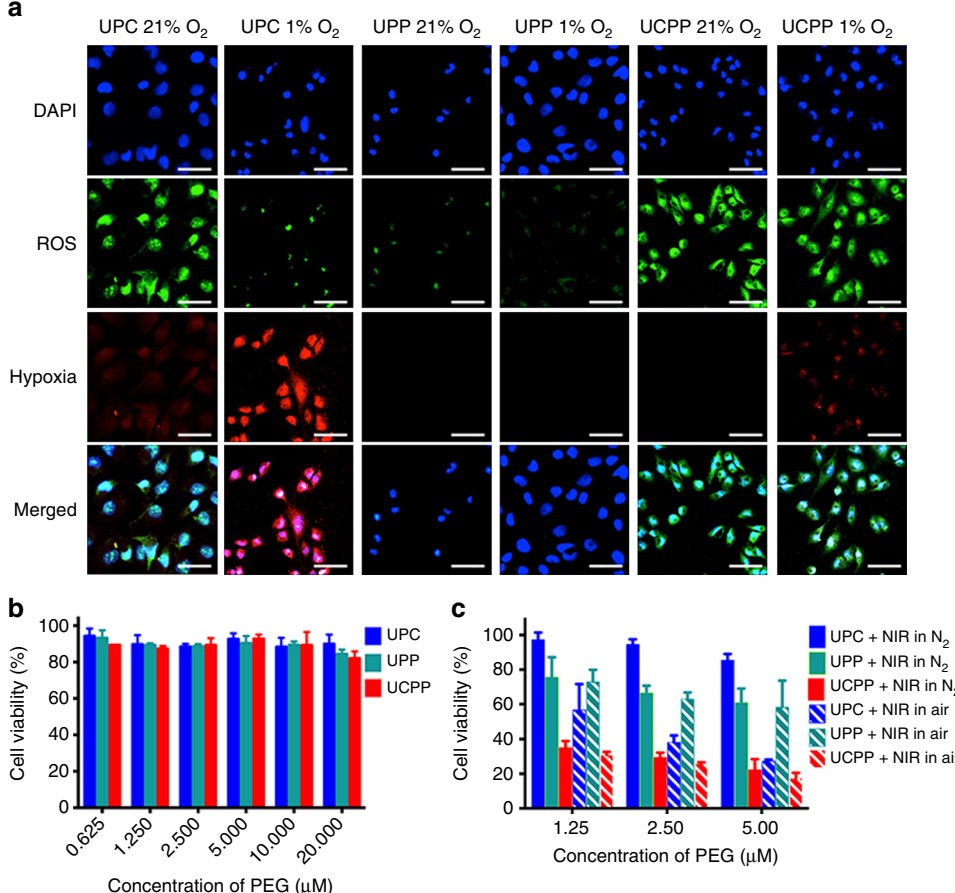

**Fig. 3** In vitro study of oxygen generating, PDT enhancement and cytotoxicity. **a** CLSM images of PDT-induced hypoxia reversion and intracellular ROS generation. The scale bars represent 20 μm. **b** Relative viabilities of L929 cells after incubation with various concentrations of UPC, UPP or UCPP without light irradiation for 48 h. **c** Relative cell viability of UPC-, UPP- or UCPP-treated HeLa cells under 980 nm light irradiation (5 min per every well) in hypoxic and normoxic environments. Data were shown as mean ± S.D. ($n = 6$).

dichlorofluorescein (DCFH) in the deoxygenated PBS. If ROS appears in the solution, the fluorescence intensity of DCFH would increase with time. In the hypoxic environment, the fluorescence intensity of UCPP-dispersed solution increased quickly (Fig. 2f). This result certified that the ROS self-generating capacity of UCPP was unaffected even in hypoxic environment upon irradiation. Meanwhile, the intensities of UPP- and UPC-dispersed solutions were extremely weak under the 980 nm irradiation, which illustrated that the ROS concentrations generated from them were extremely low. These results indicate that UCPP can reverse PDT-induced hypoxia and improve the PDT efficacy in hypoxic environment.

In order to confirm that UCPP can produce $O_2$ under NIR irradiation to reverse PDT-induced hypoxia and improve PDT efficacy, we carefully investigated this unique characteristics with L929 cells using confocal laser scanning microscopy (CLSM). As a normal cell line, the ROS level in L929 is extremely low. ROS/hypoxia detection kit was used as an intracellular hypoxia and ROS probe. The nitro group of the hypoxia detection probe can be reduced to hydroxylamine and amino groups in a hypoxic intracellular environment, inducing a red fluorescence from weak to brilliant. Meanwhile, 2′,7′-dichlorofluorescein diacetate (DCFH-DA), an intracellular ROS sensor, in the detection probe with no fluorescence can be hydrolyzed by intracellular esterase and rapidly oxidized by ROS to produce bright green fluorescent DCF. As demonstrated in Fig. 3a, under normoxic environment, UPC displayed remarkably strong green fluorescence and weak red florescence since there was high enough oxygen concentration. After NIR irradiation in a hypoxic environment, it changed to bright red fluorescence and negligible green fluorescence. In contrast, either in normoxic or hypoxic environments, UCPP-treated L929 cells showed bright green fluorescence and rather

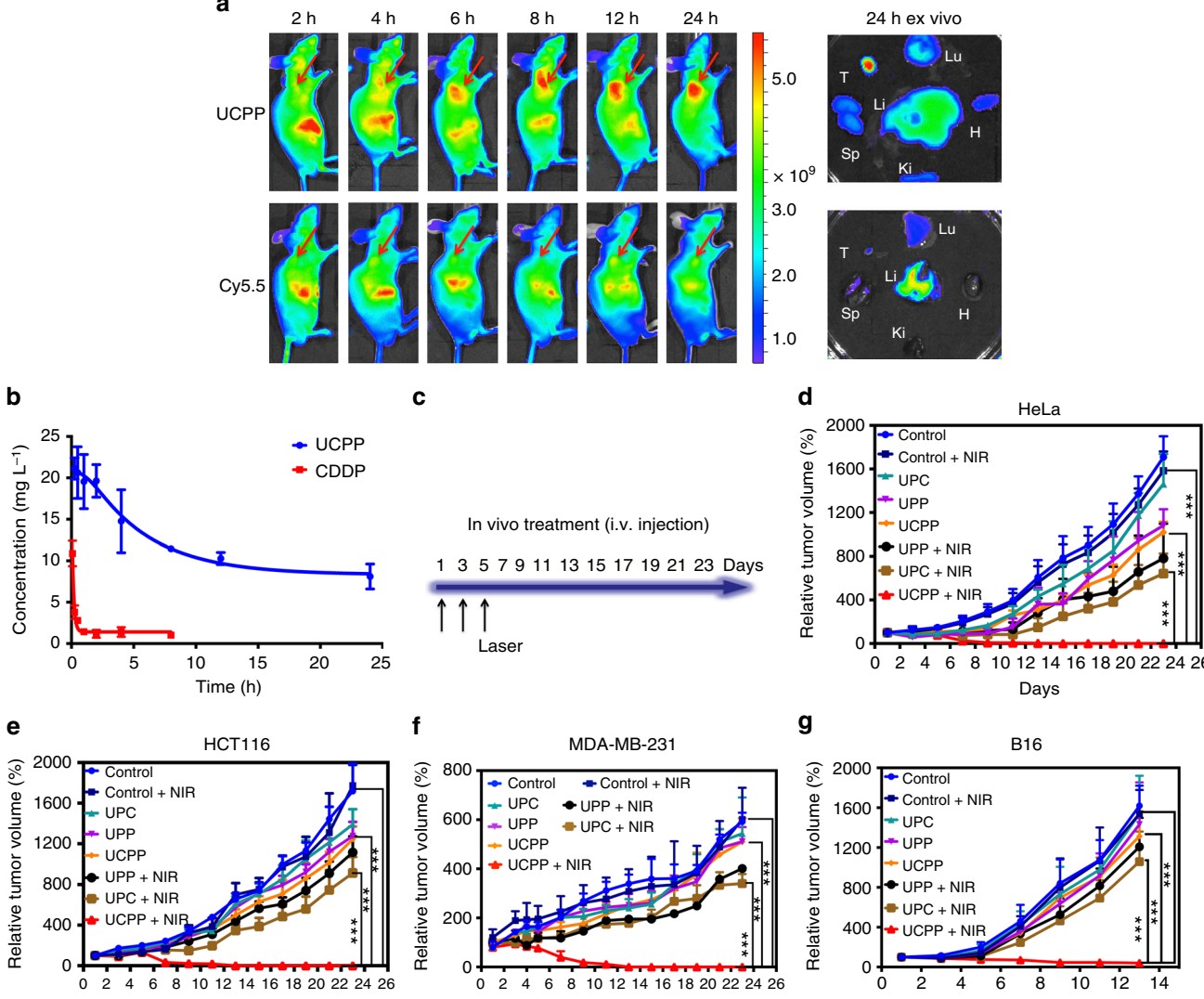

**Fig. 4** In vivo behavior and antitumor activity of UCPP. **a** In vivo fluorescence imaging of UCPP at different time points after intravenous injection. The right column shows ex vivo fluorescence images of major organs and tumor striped from those mice at 24 h after intravenous injection (red arrows indicates the locations of tumors) where T, H, Li, Sp, Lu, and Ki stand for tumor, heart, liver, spleen, lung, and kidney, respectively. **b** Profiles of plasma concentration vs. time using UCPP nanoparticles and free CDDP (a dose of 1 mg cisplatin per kg body weight). Data were shown as mean ± S.D. (n = 4). **c** Schematic illustration of in vivo antitumor treatment. **d–g** Tumor volume of HeLa, HCT116, MDA-MB-231, and B16 tumor-bearing mice after treating with PBS (blue), PBS + NIR (deep blue), UPC (cyan), UPP (purple), UCPP (orange), UPP + NIR (black), UPC + NIR (brown), and UCPP + NIR (red). Data were shown as mean ± S.D. (n = 5). P values by comparing the UCPP + NIR group with other control groups were calculated by two-tailed Student's t-test (***p < 0.001, **p < 0.01, or *p < 0.05)

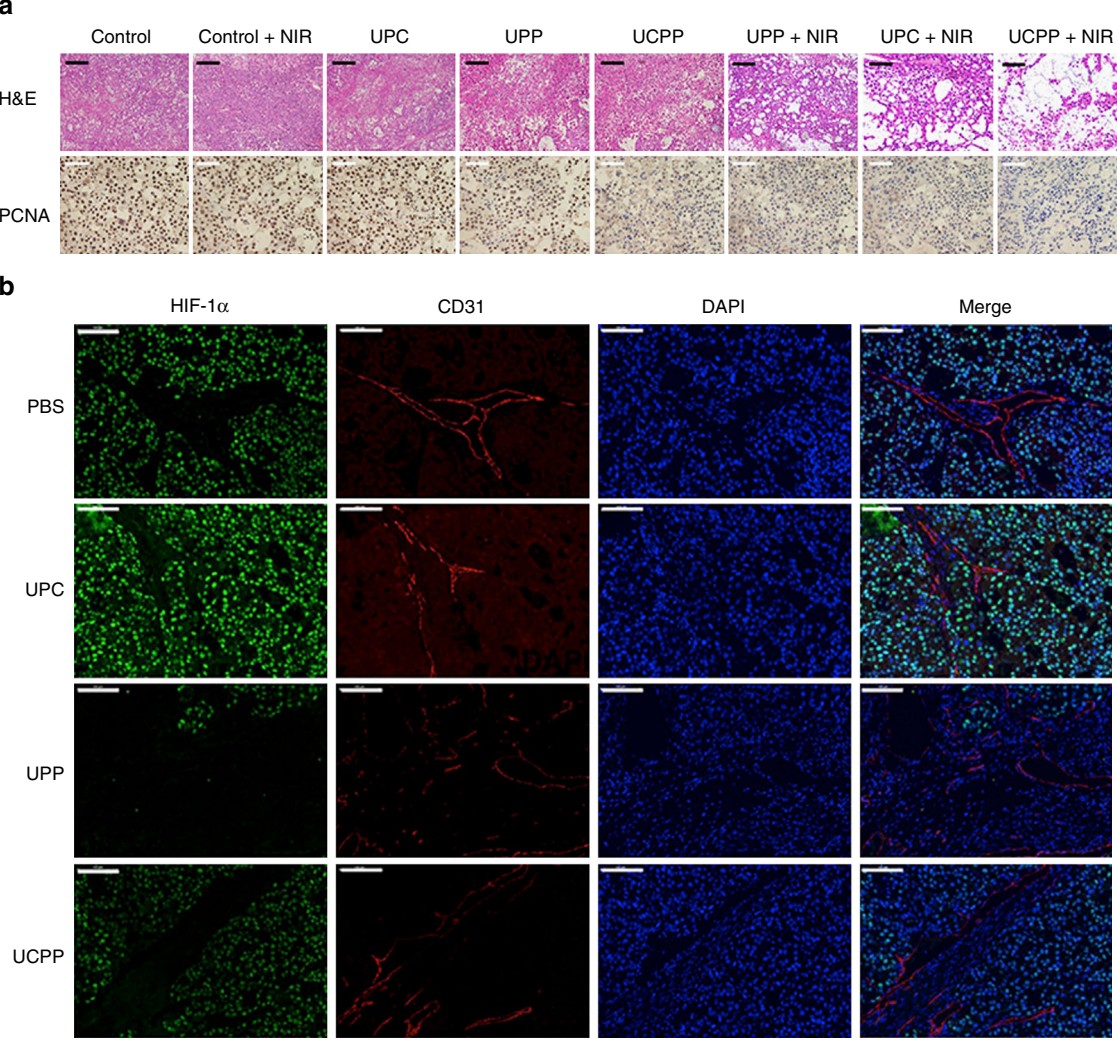

**Fig. 5** In vivo studies of anticancer efficacy and hypoxia reversion. **a** H&E and PCNA stained HeLa tumor slices from different groups collected 24 h after one single injection and light irradiation respectively (magnification × 200). The scale bars represent 100 μm. **b** Representative immunofluorescence staining of HIF-1α (green) and CD31 (red) on HeLa tumor slices collected 24 h after one single injection and light irradiation (magnification × 200). The scale bars represent 100 μm

weak red fluorescence, suggesting that UCPP can really generate oxygen inside the cells and afford the $O_2$ consumption in PDT to overcome PDT-induced hypoxia in vitro.

**Biocompatibility and cytotoxicity studies**. For a drug delivery system, biocompatibility is most important for its clinical applications. In a PDT treatment, high photo and low dark toxicity of nano-system are generally required. As displayed in Supplementary Fig. 14 and Supplementary Discussion 5, NIR irradiation alone had negligible effect towards therapeutic efficacy. At the meantime, dark toxicity of UCPP towards L929 cells was extremely low even at a concentration of 20 μM, due to its quite inert in dark, even in the existence of the reducing agent with milli-molar concentration, as shown in drug release data (Figs 2d, 3b).

Once exposed to laser irradiation, UPC showed high toxicity towards HeLa cells in normoxic environment. However, no significant cytotoxicity was observed in hypoxic condition (Fig. 3c, Supplementary Fig. 15 and Supplementary Discussion 5). This result indicated that negligible ROS was generated by UPC in hypoxic environment. In contrast, when UCPP was also exposed to laser irradiation, a similar cytotoxicity could be detected in both normoxic and hypoxic environments. This result revealed

that UCPP could self-generate oxygen to satisfy the need of ROS generation in hypoxic medium. At the meantime, active cytotoxic Pt(II) was released by UCPP under irradiation for synergistic PDT-chemotherapy. Thus, the therapeutic efficacy of UCPP is significantly higher compared to that of UPC and UPP either in normoxic or hypoxic environment.

**In vivo blood circulation and biodistribution studies**. It is well known that drugs in nanosized formulation usually exhibit a long retention time in the bloodstream, which facilitates the drug accumulation at the tumor site. As expected, on the basis of Pt quantification by ICP-MS, cisplatin (CDDP) was eliminated rapidly from circulation in the first half an hour, whereas UCPP retained in the bloodstream at a higher concentration up to 24 h (almost 38.6% remained for UCPP) (Fig. 4b). Moreover, to verify the tumor retention ability of UCPP, nanoparticle was labeled with a near-infrared fluorescence (NIRF) dye Cy5.5 for in vivo imaging. Both TEM and DLS observations (Supplementary Figs 12, 13 and Supplementary Discussion 4) showed that the size of Cy5.5-UCPP nanoparticles remained almost unchanged comparing to that of UCPP. As shown in Fig. 4a, the free Cy5.5 was rapidly cleared over the time and the fluorescence intensity at the

tumor site was extremely low. In contrast, the fluorescence signal of Cy5.5 gradually increased at the tumor site even up to 24 h when dye-labeled UCPP was used. After sacrificing the mice, ex vivo biodistribution of the drug at different organs and tumors confirmed that UCPP really accumulated at the tumor site. Tissue distribution studies, which were carried out on the base of Pt quantification using ICP-MS, further demonstrated the tumor retention ability of UCPP and indicated its potential for anticancer treatment in vivo (Supplementary Fig. 16 and Supplementary Discussion 6).

**In vivo anticancer therapy of UCPP.** To evaluate the anticancer activity of UCPP in vivo, four tumor modes were established, including two subcutaneous tumor model HeLa and HCT116, two superficial carcinoma in situ B16 and MDA-MB-231. These mice bearing HeLa, HCT116, MDA-MB-231, and B16 cells were divided into seven groups respectively and intravenously injected the UCPP or control samples once every 2 days for three times. On the 5th day, the tumors were subjected to light irradiation for 10 min (Fig. 4c). The tumor sizes and body weights were measured every 2 days. As shown in Fig. 4d–g, negligible effect on tumor growth was found when administrating UPC without laser irradiation, and chemotherapy using UPP or UCPP alone showed only a slight therapeutic efficacy to inhibit the tumor growth. Thermal imaging observations indicated that the heating effect of NIR alone is negligible and has little influence towards experimental results (Supplementary Fig. 17 and Supplementary Discussion 7). With irradiation, PDT treatment using UPC nanoparticles resulted in a delayed tumor growth inhibition in the first several days, and then the tumor continuously grew in the rest of the days, implying that the incomplete apoptosis of tumor cells and hypoxia-induced resistance of PDT may cause regrowth of the tumor. Surprisingly, all xenografted tumors disappeared only after laser irradiation without recurrence and along with little side effect on body weight when HeLa, HCT116, and MDA-MB-231 tumor-bearing mice were treated with UCPP nanoparticles and 980 nm laser irradiation (Fig. 4d–f, Supplementary Figs 19, 21, 22, 24, 26, 28, and 30). Since melanoma is the most malignant tumor in cutaneous tumors, the volumes of B16 tumors increased unimaginably fast. Even then, the therapeutic efficacy of UCPP plus with 980 nm irradiation is significantly higher than that of other control groups. Two of the five mice were healed when treated with UCPP nanoparticles and 980 nm laser irradiation, and the tumor volumes of other mice shrank to very small (Fig. 4g, Supplementary Figs 31 and 32). After different treatments, mice were sacrificed and tumors were stripped out to weigh. Attributing to the exciting therapeutic efficacy of UCPP plus with 980 nm irradiation, the weights of HeLa, HCT116, and MDA-MB-231 tumors are reduced to zero (Supplementary Figs 18, 20, 23, 25, 27, and 29). These results clearly demonstrate that the prominent antitumor effect in vivo can be achieved by the rational designed UCPP which can self-generate $O_2$ and Pt(II) under NIR irradiation for relieving tumor hypoxia and synergistic PDT-chemotherapy. Meanwhile, negligible impact on the body weight and normal organs of HeLa tumor-bearing mice were observed (Supplementary Fig. 33 and Supplementary Discussion 8), indicating the low systematic toxicity of UCPP.

Lastly, hematoxylin and eosin (H&E) stain and proliferating cell nuclear antigen (PCNA) expression were applied to HeLa tumor slices from different groups collected 24 h after one single injection and light irradiation. As shown in Fig. 5a, most addition area of fragmentation in H&E stain and least number of positive tumor cells revealed in the PCNA expression were observed in the synergistic therapy group using UCPP, while the cells in other groups largely or partly remained with their normal morphology and proliferation (Supplementary Fig. 34 and Supplementary Discussion 9). These results confirmed that the synergistic photochemotherapy using UCPP efficiently induced apoptosis and inhibited proliferation of tumor cells. As a result, it significantly enhanced therapeutic efficiency.

**In vivo overcoming hypoxia study of UCPP.** To further validate the capability of UCPP to ameliorate tumor hypoxia, HeLa tumor slices striped from the mice 24 h after a single injection of UCPP and laser irradiation were stained with fluorescent dye-labeled antibody to visualize the expression of hypoxia-inducible factor-1α (HIF-1α, green) and CD31 (red), two typical proteins that can indicate the hypoxia level in specific tissues. HIF-1α is a protein promoting blood vessel formation and degrades in the normoxic environment[28]. Hence, the decrease of HIF-1α and CD31 levels are typical symbols of hypoxia reversion[21]. Compared to the PBS group, UPC treated tumor slice displayed obviously enhanced green fluorescence, indicating that UPC exaggerate the tumor hypoxia (Fig. 5b, Supplementary Fig. 35 and Supplementary Discussion 10). On the contrary, after UPP and UCPP treatment, reduced expression of HIF-1α was observed, which is attributed to the $O_2$-generation of Pt(IV) under irradiation. Moreover, the down-regulation of HIF-1α in UCPP treated group was slightly lower than that of UPP group since the UCPP would consume part of $O_2$ generated by Pt(IV) during the PDT process. Accordingly, the blood vessel intensities in UCPP and UPP groups appeared less than that in UPC and PBS groups, indicating the alleviated and less hypoxia of the tumor microenvironment. These immunofluorescence stain results indicated that our $O_2$ self-generating drug complex would be a promising candidate to overcome the hypoxia-associated resistance in the PDT treatment.

## Discussion

In summary, a multifunctional NIR-controlled and $O_2$/Pt(II) self-generating UCPP is rationally designed and synthesized for improving PDT efficacy against hypoxic cancer and synergistic photo-chemotherapy. Under 980 nm irradiation, UCPP can self-generate $O_2$ to form ROS and reverse the hypoxia-associated PDT resistance, meanwhile release Pt(II) for chemotherapy. Owing to their effectively accumulation at the tumor site as well as the combined photodynamic and chemotherapy, significant antitumor efficacy is achieved using UCPP even at a relatively low dosage in a unitary treatment. Importantly, all tumors treated with UCPP and NIR irradiation were totally disappeared without recurrence. As a therapeutic method that self-generates $O_2$, ROS and Pt(II) species to enhance the antitumor efficacy in a synergistic way, the multifunctional UCPP presented great potential for clinical translation.

## Methods

**Materials.** Cell Counting Kit (CCK-8), PEG-NH$_2$ and NH$_2$-PEG-NH$_2$ ($M_w$ = 2000, > 99%, sigma), anhydrous dimethyl sulfoxide (DMSO), 1-ethyl-3-(3-dimethylaminopropyl) carbodiimide (EDC), N-hydroxysuccinimide (NHS), Cyanine5.5-NHS ester were purchased from Sigma Aldrich Co. (St. Louis, MO, USA) and used as received without further purification. Chlorin e6 (99%, J&K), cisplatin (CDDP, 98%, Meilunbio, Dalian), 2',7'-Dichlorofluorescin diacetate (DCFH-DA, Beyotime Ltd), ROS-ID® hypoxia/oxidative stress detection kit (Enzo Life Sciences Inc), CD31 and HIF-1α antibodies (Abcam Inc) were used as received.

**Instrument and characteristics.** The morphologies of UCPP and UCNPs were examined by transmission electronic microscopy (TEM, Tecnai G2spirit Biotwin) at an accelerating voltage of 120 kV. The platinum content obtained outside of the dialysis bags in drug release experiments was measured on Inductively Coupled Plasma Mass Spectrometer (ICP-MS, iCAP Qc-ICPMS, Thermoscientific, USA) and Inductively Coupled Plasma Optical Emission Spectrometer (ICP-OES, iCAP 6300, Thermoscientific, USA). $^1$H NMR spectra were measured by a Unity-300

MHz NMR spectrometer (Bruker) at room temperature. UV-vis absorption spectra were taken on a Milton Ray Spectronic 3000 array spectrophotometer. Photoluminescence (PL) spectra were measured on a Perkin-Elmer LS-55 spectrofluorometer. Mass Spectroscopy (ESI-MS) measurements were performed on a Quattro Premier XE system (Waters) equipped with an electrospray interface (ESI). Size and size distribution of micelles were determined by DLS (Zetasizer nano ZS, Malvern, UK). Fourier transform infrared (FTIR) spectra were recorded on a Paragon 1,000 instrument by KBr sample holder method. Oxygen concentration was measured by a oxygen dissolving meter (HI-2400, HANNA, Italy).

**Drug release**. The drug release of UCPP in dark was conducted by dialyzing (MWCO = 1000) 2 mL of UCPP (2 mg per mL) in 50 mL different buffered solutions (PBS, 0.1 mM DTT, 10 mM DTT) at 37 °C. While drug release of UCPP under NIR irradiation was conducted by exposing 2 mL UCPP (2 mg per mL) to 980 nm NIR irradiation for 5 min and then dialyzing in 50 mL different buffered solutions (PBS, 0.1 mM DTT, 10 mM DTT) at 37 °C. At predetermined time intervals, 2 mL of external buffer solution was took out and analyzed by ICP-MS.

**O$_2$ generation**. UCPP, UPP or UPC was dispersed in fresh PBS at the concentration of 8 mM (PEG). After that, 1 mL of liquid paraffin was added to isolate fresh PBS and air. The solution was exposed to 980 nm irradiation at a power density of 0.85 W cm$^{-2}$. A HI-2400 oxygen dissolving meter was used to measure the O$_2$ generation in real-time.

**ROS generation**. Firstly, DCFH-DA was transform to DCFH according to the report[29]. Briefly, 2 mL of 0.01 N NaOH was added to 0.5 mL of 1 mM DCFH-DA in ethanol and stirred at room temperature for 30 min. The solution was then neutralized with 10 mL of PBS. Then, 1 μL of DCFH solution was added to 100 μL of UCPP, UPP or UPC nanoparticles (with a concentration of 8 mM) dispersed in deoxygenated PBS. The solution was illuminated under a 980 nm irradiation at a power density of 0.85 W cm$^{-2}$. Emission intensity of various solutions was recorded with a BioTek Synergy H4 hybrid reader (fluorescence intensity was measured with excitation at 495 nm and emission at 580 nm).

**In vitro study of O$_2$ self-generation and PDT enhancement**. To further study the O$_2$ and ROS generation ability of UCPP, L929 cells were seeded in 12-well plates and incubated with UCPP, UPP or UPC (with a concentration of 8 mM) in advance. Then, these cells were exposed to 980 nm irradiation at a power density of 0.85 W cm$^{-2}$. Cyto-ID® Hypoxia/Oxidative Stress Detection kit were added to the plates and incubated for 0.5 h for confocal imaging. The fluorescence distribution of hypoxia region was measured with excitation at 590 nm and emission at 670 nm. The fluorescence distribution of ROS was measured with excitation at 490 nm and emission at 525 nm. The fluorescence of Hoechst 33342 was measured with excitation at 346 nm and emission at 460 nm.

**In vitro cytotoxicity assay in different environments**. For biocompatibility assay, L929 cells were seeded into 96-well plates (0.8 × 10$^4$ per well). After adherent overnight, several concentrations (0.625, 1.25, 2.5, 5, 10, and 20 μM) of UCPP, UPP or UPC were added and incubated for another 48 h in the dark. Later on, culture medium in every well were moved and PBS was added to wash three times. Cell viability was assessed with a standard CCK-8 method. For photodynamic therapy, HeLa cells were seeded into 96-well plates (1 × 10$^4$ per well). After adherent, these cells were incubated overnight with UCPP, UPP or UPC nanoparticles at the concentration of 0.625, 1.25, 2.5 μM (Ce6) in two oxygen environment (21 and 0 %) (Incubator placed with Anaero Pack-Anaero (Mitsubishi Gas Chemical Co. Inc., Japan) to supply cell culture environments of oxygen concentrations of 0 %.). Following, the 96-well plates were placed in air or N$_2$ and incubated in the dark or exposed to 980 nm irradiation for 5 min per every well at a power density of 0.85 W cm$^{-2}$. The hypoxia experiments were conducted in the incubator with four Anaero Pack-Anaero and full of N$_2$ at 37 °C. Firstly, the laser device placed on the cell plate and four Anaero Pack-Anaero were put in the above incubator. Secondly, the incubator was filled with N$_2$ to replace the air for 5 h. After one well was irradiated for about 5 min, we opened the door of the incubator and moved the next well of plate under the laser quickly. During the experiments, the incubator was kept the constant flow ventilation of N$_2$ to ensure the cells under hypoxic environment. Then, HeLa cells were transferred into fresh media and further incubated overnight. Later on, culture medium in every well were moved and PBS was added to wash three times. Cytotoxicity was assessed with a standard CCK-8 method.

**Animals and tumor models**. All animal experiments were approved by Animal Ethics Committee of Shanghai Jiao Tong University. SD rats (~160 g) and Balb/c nude mice (5 weeks) were supplied by Chinese Academy of Sciences (Shanghai). SD rats were used to study the pharmacokinetics of UCPP and free CDDP. Balb/c nude mice were used to study the antitumor activity of UCPP. The male nude mice were injected into the right flank subcutaneously with 200 μL of cell suspension containing 1 × 10$^6$ HeLa, B16, and HCT116 cells. The female nude mice were injected into the second left breast subcutaneously with 100 μL of cell suspension containing 1 × 10$^6$ MDA-MB-231 cells. These cells line were obtained from Shanghai Cell Bank, Chinese Academy of Sciences (CAS). BALB/c mice bearing HeLa, HCT116 and MDA-MB-231 tumors were randomly sorted for treatment after 14 days when the tumor volumes reached approximately 100 mm$^3$. BALB/c mice bearing B16 tumors were randomly sorted for treatment after 7 days when the tumor volumes reached approximately 100 mm$^3$. Tumor volume was calculated using following equation: $V = 0.5 \times A \times B^2$ ($A$ refers to the tumor length and $B$ refers to the tumor width).

**In vivo combination therapy**. Nude mice bearing subcutaneous HeLa tumors (~100 mm$^3$) were divided into eight groups randomly (5 mice per group): (a) PBS; (b) PBS and irradiated by the 980 nm light (0.8 W cm$^{-2}$ for 10 min, irradiated at 4 h p.i.); (c) i.v. injected with 200 μL UPC without irradiation; (d) i.v. injected with 200 μL UPC and irradiated by the 980 nm light (0.8 W cm$^{-2}$ for 10 min, irradiated at 4 h p.i.); (e) i.v. injected with 200 μL UPP without irradiation; (f) i.v. injected with 200 μL UPP and under the same light irradiation; (g) i.v. injected with 200 μL UCPP without irradiation; and (h) i.v. injected with 200 μL UPP and under the same light irradiation. The dose of Ce6 and $cis$-Pt(IV) were kept at 1.99 mg per kg body weight and 1.17 mg per kg body weight in these groups. Mice in c, e, and g groups were i.v. injected for three times and one time light irradiation. Mice were anaesthetized using 4% chloral hydrate through intraperitoneal injection before the irradiation. Tumor growth was measured by vernier caliper every 2 days for 3 weeks and tumor volume was calculated using following equation: $V = 0.5 \times A \times B^2$ ($A$ refers to the tumor length and $B$ refers to the tumor width). Relative tumor volume was defined as $V$ per $V_0$ ($V_0$ is the tumor volume when the treatment was initiated). Body weight was also monitored as a parameter of toxicity. Other tumor models, such as B16, HCT116 and MDA-MB-231 were conducted to the similar protocol of HeLa.

**In vivo overcoming hypoxia study**. HeLa tumor-bearing mice (~100 mm$^3$) were tail vein injected with PBS, UPP, UPC and UCPP. The dose of Ce6 and $cis$-Pt(IV) were kept at 1.99 mg per kg body weight and 1.17 mg per kg body weight in these groups. 4 h after the injection, mice were received light irradiation at the tumor sites. After 24 h post irradiation, mice were sacrificed and the tumors were collected for immunofluorescence analysis. The tumors of mice were fixed in 4% formalin for HIF-1α and CD31 staining (dilution 1:100, ab82832 and ab28364, Abcam). Nuclei of tumor cells were stained with DAPI (dilution 1:5000, Invitrogen). The images were obtained by a confocal microscopy (Zeiss LSM-710 microscope).

**Data availability**. The authors declare that the main data supporting the findings of this study are available within the article and its Supplementary Information files. Extra data are available from the corresponding author upon request.

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

## Acknowledgements
This work was financially supported by National key research and development plan of P. R. China (No. 2016YFA0201500), National Natural Science Foundation of China (51690151), National Natural Science Foundation of China (91527304). We thank Prof. Fuyou Li and Bin Shen from Fudan University for providing the important upconversion nanoparticles NaYbF$_4$: Tm@CaF$_2$. The authors are much obliged to Mrs. Yajuan Zou and Hang Wang (Instrumental Analysis Center of Shanghai Jiao Tong University) for ICP and LC-MS assay. Thanks Mr. Jiwen Qian and Dongbo Guo for help in animal experiments.

## Author contributions
D.Y. and S.X. designed the project and devised the experiments. S.X., X.Z., C.Z., W.H., Y. Z. and D.Y. were responsible for the data collection and analysis. D.Y. and S.X. dealt with the figures and prepared the main manuscript. All authors contributed to the discussions and manuscript preparation.

## Additional information

**Competing interests:** The authors declare no competing interests.

