## [Peer Review File · Nature Communications]

Reviewers' Comments:

Reviewer #1:

Remarks to the Author:

Authors describe the synthesis of a mixed nanoparticle drug delivery system to treat hypoxic tumors. In response to light, these nanocarriers release cytotoxic Pt²⁺ complexes and oxygen which is converted to ROS using an embedded photosensitizer. The combination of this photodynamic therapy (PDT) and synergistic chemotherapy is then evaluated in vitro (using HeLa Cells) and in vivo (xenografted HeLa-derived).

The multifunctional drug carrier is based on the telechelic oligoPEG that is functionalized with a photosensitizer (Ce6) and a light-sensitive Pt(IV) complex (pro-drug). The amphiphilicity of this construct drives its self-assembly to micelles. These micelles are then loaded with an upconversion nanoparticles which can turn the near infrared light to the frequency suitable for excitation of pro-drug.

Most of the characterization and experimental designs are through and results are presented logically. However, the efficacy of this treatment is only shown in one cancer model. My main concern is the applicability of these results to other cancer models and clinical translation of this results (as suggested in the paper). This is the most critical issue and the authors must prove efficacy in additional (and more clinically relevant tumor models) for the paper to be suitable for publication in the journal of Nature Communications.

Other points that should be addressed by the authors:

1) I found the nomenclature of these constructs to be confusing. The author can significantly assist the reader through a judicious choice of acronyms or their introduction through the text. As an example, in the abstract (Page 1, line 8):

“self-generating multifunctional nanocomposite (UCPP) may reverse...”

The UCPP abbreviation is not defined until Page 4, line 69. I encountered a similar situation when first coming across UPP and UCP abbreviations in Figure 2. It is up to authors how to address this point but as a suggestion, they can include a table in the paper and summarize all the constructs and their abbreviations for a quick reference.

2) Synthesis and characterization of CPP constructs:

2a. Please provide the integrals for all peaks in figure 2a and related supplementary figure. The integrals are provided only in one of the supplementary figures and are not legible due to the size. Please stack the graphs horizontally to avoid this issue.

2b. In the first step of the synthesis of CPP, authors mix 1 eq of the Pt catalyst with 1 eq of the bifunctional PEG. I am expecting that reaction will result in a thermodynamic mixture of products (2 Pt, 1Pt, and zero Pt) per PEG chain. The authors then purify the product from small molecules through dialysis. It is hard to envision that this reaction yields exclusive formation of mono-functionalized PEG. No NMR data is provided for this intermediate. Please Provide NMR spectra (with integral) and other characterization data such as ESI-MS or chromatography to characterize this mixture of products.

This is a critical issue because this intermediate is used in the next step without further purification and thus final CPPs are not entirely made of Ce6-PEG-Pt and are instead a heterogeneous mixture of many possible product combinations. This heterogeneity is in contradiction with the schematic presented in Figure 1 and may also have an impact on the self-assembly of CPPs.

3) Calculation of Ce6 concentration

Authors have characterized cell viability and toxicity under NIR for various constructs in Figure 3b and 3c. In each case, green bar graphs are corresponding to UPP constructs that lack Ce6 photosensitizer. These bars are plotted on an x-axis that is labeled Ce6 concentration. This issue is

also occurring in Figure S3. Authors need to explain how Ce6 concentration is calculated for these constructs or find an alternative way to plot this data.

Reviewer #2:

Remarks to the Author:

This manuscript reports a method to reverse PDT triggered hypoxia and treatment resistance using an upconverting nanosystem that self-generates oxygen. Besides its capacity to generate oxygen, the nanocomplex also includes a conventional photosensitizer Ce6 as well as chemotherapeutic agent Pt(II), which are activated upon irradiation with NIR light. The design of this nanocomplex is based on self-assembly of the Ce6-PEG-Pt(IV) or CPP micelles and the UCNPs, where the UCNPs get embedded in the core of the CPP micelles. Upon irradiation with NIR light at 980 nm, the core UCNs emit upconverted light at 365 nm (UV) and 470 nm (blue), which trigger the decomposition of Pt(IV) to Pt(II) and generate oxygen; as well as 660 nm light (green), which activates Ce6 leading to the production of ROS. The rationale behind the design of the nanoconstruct is good; however, the design itself might have some shortcomings due to the inability to control the loading/embedding of the UCNPs in the CPP micelles during the synthesis process. Hence, the authors might consider addressing the following suggestions that can potentially make this work more appealing.

Major corrections:

1. The TEM images of the free UCNs seem to be more or less uniform in size (around 20 nm). However, following encapsulation of the UCNPs in the CPP micelles, the size of the UCPP nanoparticles (inset of Figure 1b) does not seem to be very uniform. Does this mean that the number of UCNPs embedded within the micelles is not uniform and cannot be controlled? If this is the case, then it may lead to batch variations.
2. How do you remove the self-assembled CCP micelles without the UCNs, from the UCPP mixture?
3. What is the NIR light dose used in the solution based oxygen and ROS production experiments?
4. How many technical replicates are there for each point in Figure 1E and F? Please include error bars in these figures. Technical replicates are necessary to prove that the UCPPs have no batch variations.
5. For in vitro PDT study with both L929 cells (Figure 3c) and HeLa cells (Figure S3), NIR light alone control is missing. Since NIR light at 980 nm has a heating effect, it is likely that some cells might die simply due to NIR light alone. Duration of irradiation per well seems to be fairly long at 5 mins/well. Hence, it is necessary to include a NIR light alone control.
6. It is mentioned that each well is irradiated for about 5 mins/well. Hence during NIR irradiation was the plate maintained in hypoxic conditions during irradiation? Was the whole NIR set up in a hypoxic chamber. If the plates are left outside for too long, it might re-oxygenate the cells, especially because I guess the lid of the 12 well plate were removed during irradiation. Please clarify how the hypoxic conditions were maintained during irradiation and include the details in the methodology section.
7. Line 392: It is stated that the media was replaced with fresh medium immediately after NIR irradiation. What is the rationale behind this? Wouldn't the Pt(II) released be quickly removed before it exerts its cytotoxic effects?
8. Please include the methodology for conjugation of Cy5.5 dye to UCPP? Understand that the conjugation was done to confirm the accumulation UCPP at the tumor site. However, does the addition of a dye molecule preserve the properties of the UCPP?
9. Similarly, for in vivo PDT, the NIR light alone control is not included in figures 4d, 4f, 4g, 4h and S5. Please include this as well.
10. Figure 5 a and b: It will be better to do image analysis and quantify the staining of PCNA, HIF-1 α and CD31, so that it can be compared between groups.
11. In both pharmacokinetic and biodistribution studies, the animals were only injected once with

the nanocomplexes at a dose of 1 mg cisplatin/kg. However, in the therapeutic study, the mice were given 1.17 mg cisplatin/kg 3 times on days 1, 3 and 5 before NIR irradiation?

What is the rationale behind choosing such a regimen for therapeutic study?

12. What is the ethical tumor size limit approved by the IACUC/ animal welfare committee of your institution? The tumours of animal in Figure S5 (Control and UPC group) seems very huge.

Minor corrections:

1. There are a lot of grammatical errors throughout the manuscript. The authors will have to do a thorough check before submission of the edited version of the manuscript.
2. Line 25: What do you mean by "deformed blood tumor angiogenesis?"; should it be blood vessel deformation during tumor angiogenesis?
3. Line -35-36: References 14-20 are clubbed together. It will be better to separate the reference for the various O₂ generating materials, for the ease of referring to the particular papers.
4. Figure 3a. Add scale bars to the CLSM images and state the magnification in the figure legend.
5. Line 278 and 432: Revise the sub-title "In-vivo overcoming hypoxia study".
6. Figure 5 a and b: Add scale bars to the CLSM images and state the magnification in the figure legend.
7. Figure S6: include scale bar in H&E images. Mention the magnification in the legend. Remove the word "normal" from the legend.
8. Were the animals anaesthetised during the irradiation? I guess they will be, but there are no details on the anaesthetic used. Please include that.
9. In the methodology section, there is a lot of missing information like the company name and/or model of the equipment used (HI-2400 meter, Biotek Synergy H4 hybrid reader, NIR laser set up, Ex-vivo whole animal imaging set up, CLSM system and specs of the lasers used). This is important for anyone who wants to reproduce this work.

Reviewer #3:

Remarks to the Author:

Recommendation: It appears that the publication of this work in Nature Communication in any form would be premature at this time.

The manuscript describes a 980 nm light-controlled and O₂/Pt(II) self-generating prodrug, which potentially enhance PDT efficacy for tumor synergistic photo-chemotherapy. There are a few design flaws in this system.

1. Based on the research proposal (Figure 1), one UCPP molecule has one Pt(IV) and one Ce6 linked by short hydrophilic OEG chain, upon irradiation by a 980 nm laser, one UCPP molecule just generates one O₂ molecule, which is not enough to significantly change the O₂ state in tumors, and the O₂ generating efficiency is hard to compare many well-demonstrated MnO₂ systems.
2. For singlet oxygen test, the O₂ dissolved in the water dispersion was not excluded, whereas it is expected be much than the extra oxygen generated from nanoparticles. herein the in vitro evaluation is unable to confirm the proposal.
3. The heating effect of 980 nm is unavoidable, how to exclude the heating effect to the singlet oxygen generation and the resultant therapeutic effect?
4. Most of the NMR spectra in this work were unclear to demonstrate the structure, and other demonstration spectroscopy are also absent to confirm the exact structure.
5. In figure 2d, the time scale of drug release is so long, over 70 days, and the experimental detail concerning light irradiation is confusing in the supporting part, which make the result is hard to be

convincing.

In all, the quality of the manuscript including the data reliability and the clarity of data presentation as well as idea expression is immature for the publication in the top journal.

Reviewers' comments:

Reviewer #1:

Authors describe the synthesis of a mixed nanoparticle drug delivery system to treat hypoxic tumors. In response to light, these nanocarriers release cytotoxic Pt²⁺ complexes and oxygen which is converted to ROS using an embedded photosensitizer. The combination of this photodynamic therapy (PDT) and synergistic chemotherapy then evaluated *in vitro* (using HeLa Cells) and *in vivo* (xenografted HeLa-derived).

The multifunctional drug carrier is based on the telechelic oligoPEG that is functionalized with a photosensitizer (Ce6) and a light-sensitive Pt(IV) complex (pro-drug). The amphiphilicity of this construct drives its self-assembly to micelles. These micelles are then loaded with an upconversion nanoparticles which can turn the near infrared light to the frequency suitable for excitation of pro-drug.

Most of the characterization and experimental designs are through and results are presented logically. However, the efficacy of this treatment is only shown in one model. My main concern is the applicability of these results to other cancer models clinical translation of this results (as suggested in the paper). This is the most critical issue and the authors must prove efficacy in additional (and more clinically relevant tumor models) for the paper to be suitable for publication in the journal of Nature Communications.

Response: According to the reviewer's suggestion, other three clinically relevant cancer models were supplemented in the revised manuscript, including subcutaneous tumor model HCT116 (colorectal carcinoma), superficial carcinoma *in situ* B16 (melanin carcinoma) and MDA-MB-231 (breast carcinoma). All the results are to that of HeLa tumor model. In detail, all HCT116 and MDA-MB-231 tumors disappeared when mice were treated with UCPP nanoparticles and 980 nm laser irradiation only once. Excitingly, there is no recurrence of tumors with little side effect on body weight. Since melanoma is the most malignant tumor in cutaneous tumors, the volume of B16 tumors increased unimaginable fast. Even so, the therapeutic efficacy of UCPP plus with 980 nm irradiation is significantly higher than that of other control groups. When they were treated with UCPP nanoparticles and 980 nm laser irradiation, two of the five mice were healed and the volume of other mice's tumors shrank to very small. Here we hope to build the foundation that UCPP nanoparticles can reverse the hypoxia-associated PDT resistance to improve its anticancer efficacy significantly. In the near future, we plan to use this synergistic photo-chemo system in clinic trial to evaluate its anticancer efficacy and further work will be going on.

Supplementary Figure 23: Tumor weight of HCT116 tumor-bearing mice after treated with PBS (blue), PBS + NIR (deep blue), UPC (brown), UPP (green), UCPP (purple), UPP + NIR (orange), UPC + NIR (black), UCPP + NIR (red). Data were shown as mean \pm S.D (n = 5). P values by comparing the UCPP + NIR group with other control groups were calculated by two-tailed Student's t-test (**p < 0.01, ***p < 0.001, or *p < 0.05).

Supplementary Figure 24: Body weight of HCT116 tumor-bearing mice after treated with PBS (blue), PBS + NIR (deep blue), UPC (brown), UPP (green), UCPP (purple), UPP + NIR (orange), UPC + NIR (black), UCPP + NIR (red). Data were shown as mean \pm S.D (n = 5).

Supplementary Figure 25: Images of striped HCT116 tumors after the treatment (healed tumors were delineated with imaginary black circle).

Supplementary Figure 26: Photos of HCT116 tumor-bearing mice treated with PBS, PBS and NIR irradiation, UPC, UPP, UCPP, UPP and NIR irradiation, UPC and NIR irradiation, UCPP and NIR irradiation during 23-day treatment.

Supplementary Figure 27: Tumor weight of MDA-MB-231 tumor-bearing mice after treated with PBS (blue), PBS + NIR (deep blue), UPC (brown), UPP (green), UCPP (purple), UPP + NIR (orange), UPC + NIR (black), UCPP + NIR (red). Data were shown as mean \pm S.D (n = 5). P values by comparing the UCPP + NIR group with other control groups were calculated by two-tailed Student's t-test (**p < 0.01, ***p < 0.001, or *p < 0.05).

Supplementary Figure 28: Body weight of MDA-MB-231 tumor-bearing mice after treated with PBS (blue), PBS + NIR (deep blue), UPC (brown), UPP (green), UCPP (purple), UPP + NIR (orange), UPC + NIR (black), UCPP + NIR (red). Data were shown as mean \pm S.D (n = 5).

Supplementary Figure 29: Images of striped MDA-MB-231 tumors after the treatment (healed tumors were delineated with imaginary black circle).

Supplementary Figure 30: Photos of MDA-MB-231 tumor-bearing mice treated with PBS, PBS and NIR irradiation, UPC, UPP, UCPP, UPP and NIR irradiation, UPC and NIR irradiation, UCPP and NIR irradiation during 23-day treatment.

Supplementary Figure 31: Body weight of B16 tumor-bearing mice after treated with PBS (blue), PBS + NIR (deep blue), UPC (brown), UPP (green), UCPP (purple), UPP + NIR (orange), UPC + NIR (black), UCPP + NIR (red). Data were shown as mean \pm S.D (n = 5).

Supplementary Figure 32: Photos of B16 tumor-bearing mice treated with PBS, PBS and NIR irradiation, UPC, UPP, UCPP, UPP and NIR irradiation, UPC and NIR irradiation, UCPP and NIR irradiation during 14-day treatment.

Other points that should be addressed by the authors:

1) I found the nomenclature of these constructs to be confusing. The author can significantly assist the reader through a judicious choice of acronyms or their introduction through the text. As an example, in the abstract (Page 1, line 8): self-generating multifunctional nanocomposite (UCPP) may reverse;

The UCPP abbreviation is not defined until Page 4, line 69. I encountered a similar situation when first coming across UPP and UCP abbreviations in Figure 2. It is up to authors how to address this point but as a suggestion, they can include a table in the paper and summarize all the constructs and their abbreviations for a quick reference.

Response: We thank the reviewer for his/her careful work to help us improving the manuscript. The nomenclatures have been modified in the revised manuscript. A table has been supplemented to summarize all the constructs and their abbreviations for a quick reference to readers.

2) Synthesis and characterization of CPP constructs:

2a. Please provide the integrals for all peaks in figure 2a and related supplementary figure. The integrals are provided only in one of the supplementary figures and are not legible due to the size. Please stack the graphs horizontally to avoid this issue.

2b. In the first step of the synthesis of CPP, authors mix 1 eq of the Pt catalyst with 1 eq of the bifunctional PEG. I am expecting that reaction will result in a thermodynamic mixture of products (2 Pt, 1Pt, and zero Pt) per PEG chain. The authors then purify the product from small molecules through dialysis. It is hard to envision that this reaction yields exclusive formation of mono-functionalized PEG. No NMR data is provided for this intermediate. Please Provide NMR spectra (with integral) and other characterization data such as ESI-MS or chromatography to characterize this mixture of products.

This is a critical issue because this intermediate is used in the next step without further purification and thus final CPPs are not entirely made of Ce6-PEG-Pt and are instead a heterogeneous mixture of many possible product combinations. This heterogeneity is in contradiction with the schematic presented in Figure 1 and may also have an impact on the self-assembly of CPPs.

Response: 2a. According to the reviewer's suggestion, the integrals for all peaks have been supplemented in Figure 2a and the related supplementary Figure 1-4. These figures are provided separately for easy to read in supplementary Information.

Figure 2a. ¹H NMR spectrum of prodrug CPP.

Supplementary Figure 1: ^1H NMR spectrum of Pt(IV) prodrug.

Supplementary Figure 3: ^1H NMR spectrum of control group PEG-Ce6.

Supplementary Figure 4: ^1H NMR spectrum of control group PEG-Pt(IV).

2b. Thanks the reviewer very much for his/her carefully reviewing our manuscript. We feel sorry to make a slip of pen here. In fact, we mixed 0.1 mmol (1

equiv) Pt(IV) with 1 mmol (10 equiv, in excess) bifunctional PEG to form the intermediate, which was almost composed of 1Pt per PEG chain. During the dialysis of the intermediate in DMSO and water, it could self-assemble into nanoparticles and remained in the dialysis bag, but the excess bifunctional PEG could penetrate the dialysis bag and was removed. Furthermore, the ^1H NMR spectrum and the HPLC curve of the intermediate confirmed its chemical structure and purity.

Supplementary Figure 2: ^1H NMR spectrum of $\text{NH}_2\text{-PEG-Pt(IV)}$.

HPLC spectrum of $\text{NH}_2\text{-PEG-Pt(IV)}$.

3) Calculation of Ce6 concentration

Authors have characterized cell viability and toxicity under NIR for various constructs in Figure 3b and 3c. In each case, green bar graphs are corresponding to UPP constructs that lack Ce6 photosensitizer. These bars are plotted on an x-axis that is labeled Ce6 concentration. This issue is also occurring in Supplementary Figure 3. Authors need to explain how Ce6 concentration is calculated for these constructs or find an alternative way to plot this data.

Response: As the reviewer pointed out, UPP constructs lack Ce6 photosensitizer. But we can calculate the concentration of UPP according to the molar ratio of Pt, PEG, and Ce6 in UCPP. The molar ratios of PEG, Pt and/or Ce6 in UPP, UPC and UCPP are 1:1, 1:1 and 1:1:1, which were confirmed by ^1H NMR, UV-vis and ICP-MS measurements. Thus, according to the concentration of Ce6 in UCPP obtained from

the UV-vis standard curve (Supplementary Figure 9), we can calculate the concentration of PEG and Pt. In the revised manuscript, the concentration of PEG is plotted on an x-axis, which is more reasonable because they all contain PEG.

Supplementary Figure 9: Standard curve of Ce6.

Figure 3. (b) Relative viabilities of L929 cells after incubation with various concentrations of UPC, UPP or UCPP without light irradiation for 48 h; (c) Relative cell viability of UPC-, UPP- or UCPP-treated HeLa cells under 980 nm light irradiation (5 min/every well) in hypoxic and normoxic environments.

Supplementary Figure 15: *In vitro* cytotoxicity. (a) Relative viabilities of HeLa cells after incubation with various concentrations of UPC, UPP or UCPP with or without light irradiation in normoxic environment for 48 h. (b) Relative viabilities of HeLa cells after incubation with various concentrations of UPC, UPP or UCPP with or without light irradiation (5 min/every well) in hypoxic environment for 48 h.

Reviewer #2 (Remarks to the Author):

This manuscript reports a method to reverse PDT triggered hypoxia and treatment resistance using an upconverting nanosystem that self-generates oxygen. Besides its capacity to generate oxygen, the nanocomplex also includes a conventional photosensitizer Ce6 as well as chemotherapeutic agent Pt(II), which are activated upon irradiation with NIR light. The design of this nanocomplex is based on the self-assembly of the Ce6-PEG-Pt(IV) or CPP micelles and the UCNPs, where the UCNPs get embedded in the core of the CPP micelles. Upon irradiation with NIR light at 980 nm, the core UCNPs emit upconverted light at 365 nm (UV) and 470 nm (blue), which trigger the decomposition of Pt(IV) to Pt(II) and generate oxygen; as

well as 660 nm light (green), which activates Ce6 leading to the production of ROS. The rationale behind the design of the nanoconstruct is good; however the design itself might have some shortcomings due to the inability to control the loading/embedding of the UCNPs in the CPP micelles during the synthesis process. Hence, the authors might consider addressing the following suggestions that can potentially make this work more appealing.

Major corrections:

1. The TEM images of the free UCNs seems to be more or less uniform in size (around 20 nm). However, following encapsulation of the UCNPs in the CPP micelles, the size of the UCPP nanoparticles (inset of Figure 1b) does not seem to be very uniform. Does this mean that the number of UCNPs embedded within the micelles is not uniform and cannot be controlled? If this is the case, then it may lead to batch variations.

Response: According to the reviewer's comment, we obtained the TEM images of UCPP nanoparticles at the different concentrations. The UCPP nanoparticles seems not very uniform in original manuscript, which can be attributed to the high concentration of them. Hence, we decreased the concentration of UCPP nanoparticles and found the sizes of UCPP nanoparticles were almost uniform at the same concentration in different batches. Indeed, UCNPs loaded in CPP micelles is a random process and it is difficult to control the number of UCNPs embedded within each micelle, but the concentration of UCNPs is obeyed a certain statistic rule and similar in the different batches, which is confirmed by ICP-MS (insert in TEM images).

The Ca content (insert) and TEM images of 0.1 mg/mL UCPP with different batches. Scale bar = 0.5 μm .

2. How do you remove the self-assembled CCP micelles without the UCNPs, from the UCPP mixture?

Response: To remove the CPP micelles without the UCNPs, the solution of UCPP mixture was ultra-centrifuged (using a Beckman Coulter Avanti J 30-I centrifuge) at 100,000 rcf on a continuous sucrose gradient (30-80%) (*Nature Materials*, 2013, 12, 741-746). The layer of pure UCPP nanoparticles was recovered from the gradient with a syringe and washed twice through the 50 kDa centrifuge filter tube to remove sucrose. The residual sample was dissolved in the desired medium for characterization.

3. What is the NIR light dose used in the solution based oxygen and ROS production experiments?

Response: The NIR light dose used in the oxygen and ROS production experiments was 0.85 W cm^{-2} and added in the caption of Figure 2.

4. How many technical replicates are there for each point in Figure 1E and F? Please include error bars in these figures. Technical replicates are necessary to prove that the UCPPs have no batch variations.

Response: There are three technical replicates for each point in Figure 2e and 2f. According to reviewer's suggestion, the error bars were supplemented in Figure 2e and 2f in the revised manuscript.

Figure 2. (e) Oxygen generation or consumption of UPP, UCPP and UPC under 980 nm irradiation (0.85 W cm^{-2}); (f) The generation of ROS of UCPP, UPP or UPC under 980 nm irradiation (0.85 W cm^{-2}) in hypoxia environment, determined by the fluorescence intensity of DCFH. Data were shown as mean \pm S.D (n = 3).

5. For *in vitro* PDT study with both L929 cells (Figure 3c) and HeLa cells (Supplementary Figure 3), NIR light alone control is missing. Since NIR light at 980 nm has a heating effect, it is likely that some cells might die simple due to NIR light alone. Duration of irradiation per well seems to be fairly long at 5 mins/well. Hence, it is necessary to include a NIR light alone control.

Response: We agree with the reviewer's opinion, i.e., NIR light alone control is necessary. We used NIR light alone as the control to calculate the cell viability in MTT assay. In addition, we also measured the cell viability when HeLa cells were exposed to 980 nm irradiation for different times. The results in Supplementary Figure 14 exhibit the cytotoxicity of NIR light alone is negligible towards HeLa cells.

Supplementary Figure 14: Cell viability after being irradiated with NIR light under different irradiation times. Error bars indicate standard deviations, N = 5.

6. It is mentioned that each well is irradiated for about 5 mins/ well. Hence during NIR irradiation was the plate maintained in hypoxic conditions during irradiation? Was the whole NIR set up was in a hypoxic chamber. If the plates are left outside for too long, it might re-oxygenate the cells, especially because I guess the lid of the 12 well plate were removed during irradiation. Please clarify how the hypoxic conditions were maintained during irradiation and include the details in the methodology section.

Response: The hypoxia experiments were conducted in the incubator with four Anaero Pack-Anaero and full of N_2 at 37 °C. Firstly, the laser device was placed on the cell plate and four Anaero Pack-Anaero were put in the above incubator. Secondly, the incubator was filled with N_2 to replace the air for five hours. After one well was irradiated for about 5 min, we opened the door of the incubator and moved the next well of plate under the laser quickly. During the experiments, the incubator was kept the constant flow ventilation of N_2 to ensure the cells under hypoxic environment. The details were provided in the methodology section.

7. Line 392: It is stated that the media was replaced with fresh medium immediately after NIR irradiation. What is the rationale behind this? Would the Pt(II) released be quickly removed before it exerts its cytotoxic effects?

Response: The NPs are difficult to be totally taken in by HeLa cells *in vitro*. The residual NPs in media under NIR irradiation could also generate singlet oxygen and release Pt(IV) to kill tumor cells. Hence, we replaced the media with fresh medium to ensure the cytotoxicity was attributed to the UCPP nanoparticles accumulated in tumor cells.

8. Please include the methodology for conjugation of Cy5.5 dye to UCPP? Understand that the conjugation was done to confirm the accumulation UCPP at the tumor site. However does the addition of a dye molecule preserve the properties of the UCPP?

Response: Here Cy5.5-NHS ester was used to react with the hydroxyl of Pt(IV) in UCPP to produce the conjugates easily (provided in the Supplementary Information).

The conjugates of Cy5.5-UCPP nanoparticles were characterized by DLS and TEM measurements. The results in Supplementary Figure 12 and 13 display that both size and polydispersity index of them are similar to those of UCPP nanoparticles.

Supplementary Figure 12: DLS measurement of Cy5.5-UCPP nanoparticles.

Supplementary Figure 13: The TEM image of Cy5.5-UCPP nanoparticles (scale bar is 100 nm).

9. Similarly, for *in vivo* PDT, the NIR light alone control is not included in figures 4d, 4f, 4g, 4h and S5. Please include this as well.

Response: According to the reviewer’s suggestion, the NIR light alone control have been added in these Figures.

Figure 4d. Tumor volume of HeLa tumor-bearing mice after treating with PBS (blue), PBS + NIR (deep blue), UPC (brown), UPP (green), UCPP (purple), UPP + NIR (orange), UPC + NIR (black), UCPP + NIR (red). Data were shown as mean \pm S.D (n = 5). P values by comparing the UCPP + NIR group with other control groups were calculated by two-tailed Student’s t-test (***p < 0.001, **p < 0.01, or *p < 0.05).

Supplementary Figure 18: Tumor weight of HeLa tumor-bearing mice after treating with PBS (blue), PBS + NIR (deep blue), UPC (brown), UPP (green), UCPP (purple), UPP + NIR (orange), UPC + NIR (black), UCPP + NIR (red). Data were shown as mean \pm S.D (n = 5). P values by comparing the UCPP + NIR group with other control groups were calculated by two-tailed Student's t-test (***) $p < 0.001$, ** $p < 0.01$, or * $p < 0.05$.

Supplementary Figure 19: Body weight of HeLa tumor-bearing mice after treating with PBS (blue), PBS + NIR (deep blue), UPC (brown), UPP (green), UCPP (purple), UPP + NIR (orange), UPC + NIR (black), UCPP + NIR (red). Data were shown as mean \pm S.D (n = 5).

Supplementary Figure 20: Images of striped HeLa tumors after the treatment (healed tumors were delineated with imaginary black circle)

Supplementary Figure 21: Photos of HeLa tumor-bearing mice treated with PBS, PBS + NIR, UPC, UPP, UCPP, UPP and NIR irradiation, UPC and NIR irradiation, UCPP and NIR irradiation during 23-day treatment.

10. Figure 5 a and b: It will be better to do image analysis and quantify the staining of PCNA, HIF-1a and CD31, so that it can be compared between groups.

Response: According to the reviewer's suggestion, we adopted ImageJ software to do image analysis and quantify the staining of PCNA, HIF-1a and CD31 (Supplementary Figure 34 and 35).

Supplementary Figure 34: The relative PCNA positive areas as recorded from more than five images for each group using the ImageJ software.

Supplementary Figure 35: The relative hypoxia positive areas and blood vessel densities as recorded from more than five images for each group using the ImageJ software.

11. In both pharmacokinetic and biodistribution studies, the animals were only injected once with the nanocomplexes at a dose of 1 mg cisplatin/kg. However, in the therapeutic study, the mice were given 1.17 mg cisplatin/kg 3 times on days 1, 3 and 5 before NIR irradiation? What is the rationale behind choosing such a regimen for therapeutic study?

Response: In the therapeutic study, a dose of 1.17 mg cis-Pt(IV)/kg was adopted because the control group we used was PEG-Pt(IV), which was different from that in both pharmacokinetic and biodistribution studies. According to the results of pharmacokinetic and biodistribution studies, the fluorescence intensity of Cy5.5 gradually increased at the tumor site even up to 24 h (Figure 4a), indicating that UCPP nanoparticles could accumulate at the tumor site constantly. Hence, we expect much more nanoparticles could accumulate in the tumor tissue to enhance therapeutic efficiency through injecting three times before NIR irradiation.

12. What is the ethical tumor size limit approved by the IACUC/ animal welfare committee of your institution? The tumours of animal in Supplementary Figure 5 (Control and UPC group) seems very huge.

Response: We appreciate the reviewer's mercy heart. We consulted the IACUC/animal welfare committee in our university. The ethical tumor size limit approved by them is 1600 mm³. None of the tumor volumes *in vivo* experiments exceeded this size limit.

Minor corrections:

1. There are a lot of grammatical errors throughout the manuscript. The authors will have to do a thorough check before submission of the edited version of the manuscript.

Response: According to the reviewer's suggestion, we did our best to revise our manuscript carefully and correct the grammatical errors.

2. Line 25: What do you mean by deformed blood tumor angiogenesis? should it be blood vessel deformation during tumor angiogenesis?

Response: Thanks for the reviewer's professional suggestion. In introduction, 'deformed blood tumor angiogenesis' has been corrected into 'blood vessel deformation during tumor angiogenesis'.

3. Line 35-36: References 14-20 are clubbed together. It will be better to separate the reference for the various O₂ generating materials, for the ease of referring to the particular papers.

Response: According to reviewer's suggestion, the references have been separated for the various O₂ generating materials in the revised manuscript.

4. Figure 3a. Add scale bars to the CLSM images and state the magnification in the figure legend.

Response: The scale bars have been added to the CLSM images and the magnification has been stated in the figure caption.

Figure 3a. CLSM images of PDT-induced hypoxia reversion and intracellular ROS generation. The scale bars represent 20 μ m.

5. Line 278 and 432: Revise the sub-title; In-vivo overcoming hypoxia study.

Response: The sub-title has been corrected as '*In-vivo* overcoming hypoxia study'.

6. Figure 5 a and b: Add scale bars to the CLSM images and state the magnification in the figure legend.

Response: The scale bars and the magnification have been added into Figure 5a and b.

Figure 5. *In vivo* studies of anticancer efficacy and hypoxia reversion. (a) H&E and PCNA stained HeLa tumor slices from different groups collected 24 h after one single injection and light irradiation respectively (magnification $\times 200$). The scale bars represent 100 μm ; (b) Representative immunofluorescence staining of HIF-1 α (green) and CD31 (red) on HeLa tumor slices collected 24 h after one single injection and light irradiation (magnification $\times 200$). The scale bars represent 100 μm .

7. Supplementary Figure 6: include scale bar in H&E images. Mention the magnification in the legend. Remove the word normal from the legend.

Response: The scale bars and the magnification have been added into Supplementary Figure 6.

Supplementary Figure 33: H&E stain of organs (HeLa tumor-bearing mice) after administrated with CDDP and UCPP irradiation (magnification $\times 200$). Scale bar = 100 μm .

8. Were the animals anaesthetised during the irradiation? I guess they will be, but there are no details on the anaesthetic used. Please include that.

Response: As the reviewer's guess, the animals were anaesthetized indeed during the irradiation. The details have been added into the methods. 'Mice were anaesthetized using 4 % chloral hydrate (200 μ L) through intraperitoneal injection before the irradiation.'

9. In the methodology section, there is a lot of missing information like the company name and/or model of the equipment used (HI-2400 meter, Biotek Synergy H4 hybrid reader, NIR laser set up, Ex-vivo whole animal imaging set up, CLSM system and specs of the lasers used). This is important for anyone who wants to reproduce this work.

Response: According to the reviewer's suggestion, we have added the missing information in the methodology section. The fluorescence intensity in ROS generation experiment was measured with excitation at 495 nm and emission at 580 nm using Biotek Synergy H4 hybrid reader. The fluorescence distribution was measured at 2, 4, 6, 8, 12 and 24 h using an *in vivo* and *ex vivo* imaging system with excitation at 690 nm and emission at 700 nm. As to the CLSM system, the fluorescence distribution of hypoxia region was measured with excitation at 590 nm and emission at 670 nm. The fluorescence distribution of ROS was measured with excitation at 490 nm and emission at 525 nm. The fluorescence of Hoechst 33342 was measured with excitation at 346 nm and emission at 460 nm.

Materials. Cell Counting Kit (CCK-8), PEG-NH₂ and NH₂-PEG-NH₂ (Mw = 2000, > 99%, sigma), anhydrous dimethyl sulfoxide (DMSO), 1-ethyl-3(3-dimethylaminopropyl) carbodiimide (EDC), N-hydroxysuccinimide (NHS), Cyanine5.5-NHS ester were purchased from Sigma Aldrich Co. (St. Louis, MO, USA) and used as received without further purification. Chlorin e6 (99%, J&K), cisplatin (CDDP, 98%, Meilunbio, Dalian), 2',7'-Dichlorofluorescein diacetate (DCFH-DA, Beyotime Ltd), ROS-ID® hypoxia/oxidative stress detection kit (Enzo Life Sciences Inc), CD31 and HIF- α antibodies (Abcam Inc) were used as received.

Instrument and Characteristics. The morphologies of UCPP and UCNPs nanoparticles were examined by transmission electronic microscopy (TEM, Tecnai G2spirit Biotwin) at an accelerating voltage of 120 kV. The platinum content obtained outside of the dialysis bags in drug release experiments was measured on Inductively Coupled Plasma Mass Spectrometer (ICP-MS, iCAP Qc-ICPMS, ThermoScientific, USA) and Inductively Coupled Plasma Optical Emission Spectrometer (ICP-OES, iCAP 6300, ThermoScientific, USA). ¹H NMR spectra were measured by a Unity-300 MHz NMR spectrometer (Bruker) at room temperature. UV-vis absorption spectra were taken on a Milton Ray Spectronic 3000 array spectrophotometer. Photoluminescence (PL) spectra were measured on a Perkin-Elmer LS-55 spectro-fluorometer. Mass Spectroscopy (ESI-MS) measurements were performed on a Quattro Premier XE system (Waters) equipped with an electrospray interface (ESI). Size and size distribution of micelles were determined by DLS (Zetasizer nano ZS, Malvern, UK). Fourier transform infrared (FTIR) spectra were recorded on a Paragon

1,000 instrument by KBr sample holder method. Oxygen concentration was measured by a oxygen dissolving meter (HI-2400, HANNA, Italy).”

Reviewer #3 (Remarks to the Author):

Recommendation: It appears that the publication of this work in Nature Communication in any form would be premature at this time.

The manuscript describes a 980 nm light-controlled and O₂/Pt(II) self-generating prodrug, which potentially enhance PDT efficacy for tumor synergistic photo-chemotherapy. There are a few design flaws in this system.

1. Based on the research proposal (Figure 1), one UCPP molecule has one Pt(IV) and one Ce6 linked by short hydrophilic OEG chain, upon irradiation by a 980 nm laser, one UCPP molecule just generates one O₂ molecule, which is not enough to significantly change the O₂ state in tumors, and the O₂ generating efficiency is hard to compare many well-demonstrated MnO₂ systems.

Response: As pointed by the reviewer, the O₂ generating efficiency was limited by the concentration of UCPP, but the oxygen concentration generating from UCPP in deoxygenated PBS could afford the consumption of PDT and alleviate the hypoxia environment. More important, UCPP can self-generate oxygen and Pt(II) to overcome the hypoxia-triggered PDT resistance and significantly improve anticancer efficacy by the synergistic PDT-chemotherapy.

2. For singlet oxygen test, the O₂ dissolved in the water dispersion was not excluded, whereas it is expected be much than the extra oxygen generated from nanoparticles. Herein the *in vitro* evaluation is unable to confirm the proposal.

Response: In the tests of ROS generation and *in vitro* study of O₂ self-generation and PDT enhancement, the O₂ dissolved in the water dispersion was excluded by the different methods. When ROS concentration in PBS was tested, the buffer was deoxygenated using sodium hydrosulfite and the residual oxygen concentration was 0.500 ± 0.020 mg/L detected by HI-2400 oxygen dissolving meter. Such low oxygen concentration for ROS generation test can be considered as hypoxia condition. When ROS generation in cells was measured, the cell plates containing HeLa cells and culture medium were placed in an airtight box with Anaero Pack-Anaero overnight to form a hypoxia environment. The residual oxygen concentration in the culture medium was only 0.625 ± 0.175 mg/L detected by HI-2400 oxygen dissolving meter. Herein, we are sure that the *in vitro* evaluation is able to confirm our results.

3. The heating effect of 980 nm is unavoidable, how to exclude the heating effect to the singlet oxygen generation and the resultant therapeutic effect?

Response: We agreed with the reviewer's opinion that it is difficult to avoid the heating effect of 980 nm laser completely in the experiments of ROS generation and *in vitro* and *in vivo* therapeutic effect. So we studied ROS generation using Cyto-ID[®] Oxidative Stress Detection kit under the NIR irradiation alone. The result of CLSM exhibited that the heating effect of 980 nm laser could be negligible here.

The CLSM images of ROS generation when L929 cells exposed to NIR irradiation alone.

In addition, we also studied the heating effect of 980 nm laser on *in vitro* and *in vivo* therapeutic effect. When HeLa cells were exposed to the NIR irradiation alone for different times, the cell viability was above 90%. Furthermore, the tumor volume of HeLa tumor-bearing mice after treating with NIR irradiation alone was similar to the result without NIR irradiation. Thus, we considered that the *in vitro* and *in vivo* heating effect of 980 nm laser could be negligible here.

Supplementary Figure 14: Cell viability after being irradiated with NIR light under different irradiation times. Error bars indicate standard deviations, N = 5.

Figure 4d. Tumor volume of HeLa tumor-bearing mice after treating with PBS (blue), PBS + NIR (deep blue), UPC (brown), UPP (green), UCPP (purple), UPP + NIR (orange), UPC + NIR (black), UCPP + NIR (red). Data were shown as mean \pm S.D (n = 5). P values by comparing the UCPP + NIR group with other control groups were calculated by two-tailed Student's t-test (***) $p < 0.001$, ** $p < 0.01$, or * $p < 0.05$).

We further studied the *in vivo* heating effect of 980 nm laser alone by using the infrared thermal camera. With the increase of NIR irradiation time, the temperature of tumor site did not exceed 37 °C. This indicated that the heating effect of 980 nm laser on the experimental results is negligible.

Supplementary Figure 17: The temperature of tumor site recorded by an IR camera under NIR irradiation for different times.

4. Most of the NMR spectra in this work were unclear to demonstrate the structure, and other demonstration spectroscopy are also absent to confirm the exact structure.

Response: According to the reviewer's suggestion, the integrals for all peaks in NMR spectra have been supplemented in Figure 2a and the related supplementary Figure 1-4. These figures are provided separately for easy to read in supplementary Information. Other spectroscopies (including FTIR, XPS and UV-visible spectra) were also provided in supplementary Information to confirm the exact structure.

Figure 2a. ¹H NMR spectrum of prodrug CPP.

Supplementary Figure 1: ^1H NMR spectrum of Pt(IV) prodrug.

Supplementary Figure 2: ^1H NMR spectrum of NH_2 -PEG-Pt(IV) prodrug.

Supplementary Figure 3: ^1H NMR spectrum of control group PEG-Ce6.

Supplementary Figure 4: ^1H NMR spectrum of control group PEG-Pt(IV).

Supplementary Figure 5: FTIR spectra of PEG-Pt(IV), PEG-Ce6, and Ce6-PEG-Pt(IV).

Supplementary Figure 6: FTIR spectra of Pt(IV), Ce6, and PEG.

Supplementary Figure 7: High-resolution XPS spectra (Pt 4f) of CDDP and UCPP.

Supplementary Figure 8: UV/Vis spectra of Ce6-PEG-Pt(IV) and PEG-Ce6 in DMSO.

5. In figure 2d, the time scale of drug release is so long, over 70 days, and the experimental detail concerning light irradiation is confusing in the supporting part, which make the result is hard to be convincing.

Response: We feel sorry to make a mistake here. In fact, the unit of time scale in drug release experiments should be hour. The total time of the drug release experiments was 72 h. Figure 2d has been corrected in the revised manuscript. The experimental details concerning light irradiation have been supplemented in the methods in the revised manuscript.

In all, the quality of the manuscript including the data reliability and the clarity of data presentation as well as idea expression is immature for the publication in the top journal.

Response: All the data in our manuscript were repeated for many times, especially the *in vitro* and *in vivo* experiments. The data presentation and idea expression have been improved as best as we can according to the reviewer's suggestion. Hence, we expect our revised manuscript can meet the requirements of this top journal.

We have revised and improved our manuscript according to the Reviewers' suggestions. Thanks again for your favorable consideration.

With best regards

Sincerely yours
Dr. Deyue Yan

Reviewers' Comments:

Reviewer #1:

Remarks to the Author:

The authors have addressed all the issues raised in the previous review. These are:
Performed an anti-tumor efficacy study in additional three tumor models: subcutaneous tumor model HCT116 (colorectal carcinoma), superficial carcinoma in situ B16 (melanin carcinoma) and MDA-MB-231 (breast carcinoma). The anti-tumor effect in B16 model is not as impressive as in other models but it is significant considering it is an aggressive tumor.

Provided a table to make their nomenclature clear.

Provided all the NMR with integral ratios and re-confirmed the structure with FTIR and XPS. They have also provided the HPLC of the intermediate. They described how they calculated the concentration of the photosensitizer (Ce6) in each of their constructs and now plotted the data w.r.t PEG which is present in each construct and thus makes more sense.

The manuscript has been substantially improved and is now ready to be published in Nature Communications.

Reviewer #2:

In comments to the editor the reviewer felt the changes made by the authors addressed the previous concerns and recommended publication.

Reviewer #3:

Remarks to the Author:

The authors made some efforts to improve the clarity of the manuscript, but the novelty and inherent shortcomings still exist. The property of UV-responsive oxygen formation from Pt prodrugs was reported before, in addition, many similar systems with functions of photodynamic therapy and chemotherapy were also demonstrated in the past work, no matter the limited oxygen generation potency in this work. One Pt-PEG-Ce6 conjugate can only generate one oxygen molecule. It means the oxygen generation potency in the final UCNP-loaded UCPP nanoparticles is restricted, the potential oxygen loading content is low, the relative inorganic component is high, which is unfavorable. Given the top standard of Nature Communication journal, but current design is not a significant breakthrough, we are sorry to insist our previous decision and more professional oncology journals are suggested to be considered. No further consideration in Nature Communication is advised.